# Measurement Techniques for Low-Concentration Tritium Radiation in Water: Review and Prospects

**DOI:** 10.3390/s24175722

**Published:** 2024-09-03

**Authors:** Junxiang Mao, Ling Chen, Wenming Xia, Junjun Gong, Junjun Chen, Chengqiang Liang

**Affiliations:** College of Nuclear Science and Technology, Naval University of Engineering, Wuhan 430033, China; 33932019004@nue.edu.cn (J.M.); dxaw110@sina.com (J.G.); tank6230@sina.com (J.C.); 15527273369@163.com (C.L.)

**Keywords:** tritium, low-level beta measurement, water measurement

## Abstract

Tritium (^3^H) is one of the most critical nuclides for environmental monitoring, yet it is challenging to measure. Its high natural mobility and its potential to enter the human body through the food chain underscore the importance of not overlooking the radiation safety risks associated with tritium. The need for the online measurement of tritium at low concentrations is becoming increasingly apparent. This review examines the two principal stages of current measurement methodologies: sample preparation and radiation signal detection. It provides a summary of the tritium sample preparation and detection techniques, highlighting advances in the research with potential applications in online monitoring. The review concludes with an analysis of the issues inherent in the current techniques and offers perspectives on possible technological enhancements and future trajectories for the development of online monitoring systems for trace tritium levels.

## 1. Introduction

With the application of nuclear power technology, nuclear power has played an accelerating role in the development of society. However, since the Fukushima nuclear accident in 2011, a large amount of nuclear effluent has been generated every year, and due to the limited storage capacity for nuclear effluent at Japan’s Tokyo Electric Power Company (TEPCO), it was decided to start discharging nuclear effluent into the sea in 2023. This action places high demand on radionuclide monitoring of the surrounding natural environment. Tritium is an important component of nuclear wastewater and is one of the nuclides that is difficult to monitor, and if it enters the human body, it can cause damage to internal organs and even the risk of cancer. Tritium (^3^H) is an important isotope of hydrogen (H). The nucleus of a tritium atom consists of one proton and two neutrons. Tritium is radioactive and exists in nature in trace quantities, with a half-life of 12.32 years. When it decays, it emits β-rays and an antineutrino to produce helium (He). The β-rays have an average energy of 5.7 keV and a maximum energy of 18.6 keV [1]. The nuclide properties of low-energy beta nuclides make the measurements themselves difficult, involving certain requirements in terms of the radiation measurement sensors and the measurement methods.

Another reason for the difficulty of measurement: tritium is found in very low concentrations in the natural environment. To determine whether or not there is tritium contamination in a particular body of water, long-term monitoring data are needed for comparison. The main sources of tritium are divided into two parts: naturally occurring tritium and artificially produced tritium. Naturally occurring tritium comes from two sources. One source is cosmic rays (high-energy particles ^14^N and ^16^O) interacting with the atmosphere, leading to a nuclear reaction that forms tritium. The tritium production pathway includes the following nuclear reactions (A2)–(A5) [2] in Section A.1. Tritium initially exists in a gaseous form and then combines with oxygen in the stratosphere. Through a molecular exchange mechanism, tritium eventually enters the lower atmosphere as triturated water (HTO) [3]. The conversion process can be represented by Equation (A6) in Section A.1.

The other natural source is crustal rocks. However, the amount of tritium contributed by this pathway is extremely small. See Section A.2 for a schematic diagram of specific tritium transport in nature.

The amount of artificial tritium is higher than naturally occurring tritium. Artificial tritium is primarily derived from reactor generation and the legacy of nuclear tests. With the development of today’s nuclear technology, the number of nuclear facilities around the world is increasing. These include nuclear power plants, nuclear reactors, and nuclear fuel production plants. Tritium is mainly produced in these facilities, which can also be produced as a byproduct of nuclear tests or from nuclear waste. The production of tritium varies depending on the type of nuclear reactor, its operation, design, and structure. Heavy water reactors produce the highest amount of tritium, followed by boiling water reactors [4]. The nuclear units involved in the Fukushima accident were boiling water reactors, which also means that the nuclear effluent from the accident may contain a sizable amount of tritium.

Tritium, when released into the ocean, becomes part of the water cycle and can enter the human body. Recent studies have indicated that tritium poses a significant health risk. This research addresses the need to understand and mitigate these risks. Tritium has a range of only 6 mm in air and 0.6 μm in human tissues, and the external irradiation exposure from tritium to the human body is negligible. Tritium, after passing through the natural cycle, often exists in the form of deuterated water (HTO) or organic tritium (OBT). It enters the human body through water or food. Approximately 80% of the tritium atoms in OBT are irreversibly bound to human cells and tissues [5]. Calculations show that the effective dose rates of β-radiation from deuterated water to ingested HTO and ingested OBT are 1.8 × 10^−11^ Sv/Bq and 4.2 × 10^−11^ Sv/Bq, respectively [6]. Consuming tritium-contaminated food may result in very low doses of radiation, which nonetheless poses a risk that should be mitigated. Nuclear safety authorities believe that tritium remains a potential public health hazard. Tauchi et al. suggest that although the biological impact of HTO is minor, the stochastic effects of tritium radiation merit attention, supported by the linear no-threshold (LNT) model [7]. This model posits that even low radiation doses can increase the likelihood of stochastic effects. Consequently, establishing LNT-based measurement systems is crucial for environmental and radiological studies. Furthermore, research indicates that organic tritium compounds (OBTs), formed within the body, inflict greater harm than HTO [8]. These OBTs, utilized in cellular metabolism, lead to internal irradiation and pose risks such as cell death, chromosomal alterations, mutations, and potentially cancer. Additionally, tritium decay into helium-3 (3He) can disrupt water structures in organisms, adding to the biological impact and chemical toxicity [9].

The data on tritium restriction levels in water are summarized in Table 1 and were obtained from the following organizations: The World Health Organization (WHO), the European Commission (EC), Russia, the U.S. Environmental Protection Agency (EPA), the Ministry of Ecology and Environment of the PRC, and Canada (Ontario) and the Ontario Drinking Water Quality and Testing Standards Advisory Committee (ODWAC). See the summary table in Section A.3 for the limit values for tritium in drinking water.

Changes in tritium levels in the marine environment due to nuclear effluent discharge are likely to be at an elevated level in the future, placing demands on low tritium concentration measurements. Most nuclear power plants release tritium into the environment through controlled releases, using liquid or gaseous forms, in compliance with national government regulations. TEPCO announced in April 2021 that it planned to discharge 1.25 million cubic meters (containing about 870 TBq) of deuterated water (HTO) into the Pacific Ocean [10] and, on 24 August 2023, TEPCO officially began the discharge of nuclear effluent into the sea, with large quantities of tritium leaking into the natural environment. Although relevant laws and regulations will be established in the future to regulate dumping behavior, in the short term there may be emission measures that follow the example of Japan, which has drawn attention to the emission of nuclides [11]. 

**Table 1 sensors-24-05722-t001:** Parameter symbol overview.

Parameter Name	Parameter Meaning	Reference
MDA	Minimum detectable activity (Bq/L)	[12]
*T_1_*	Total counting time (in minutes)	
*B*	Background count rate (in minutes)	
$V_{s}$	Sample volume (in liters)	
ε	Counting efficiency	
*β_H/D_*	Separation coefficients (H and D)	
*β_H/T_*	Separation coefficients (H and T)	
*k_B_*	Boltzmann constant	[13]
*T_0_*	Thermodynamic temperature	
*ΔZPE_HD_*	Zero-point vibration energy difference (H and D)	
*ΔZPE_HT_*	Zero-point vibration energy difference (H and T)	
FOM	Figure of merit	[14]

This review offers a concise overview of emerging technologies with the potential to enable real-time monitoring of tritium in aquatic environments. Recognizing the current gap in the literature regarding the optimal duration for such monitoring, this study specifically examines a 5 h monitoring window. The primary focus is on the critical components of the measurement process, namely the preprocessing techniques and detection methodologies, as highlighted in Figure 1. The main technical challenge lies in the accurate and rapid measurement of low tritium concentrations in water, which involves two key steps: pretreatment and measurement. These steps are pivotal, as they directly or indirectly influence the measurement outcomes. Given the scarcity of references on techniques that may facilitate future online measurements, this review delves into recent technological advances in pretreatment and detection for online monitoring of low tritium concentrations. It elucidates the underlying principles, mechanisms, and material approaches that can enhance sensor detection capabilities. Furthermore, it identifies key areas for future research and anticipates emerging trends in the field.

The significance of this review is underscored by its evaluation of the current state of practical applications in low-concentration tritium monitoring and its exploration of the potential for advancements in monitoring practices. By offering insights into innovative strategies for online monitoring of low-concentration tritium, this review aims to inform and guide future research and development efforts. The ultimate goal is to enhance the capacity for real-time, low-concentration tritium monitoring, thereby improving environmental safety and radiological research.

## 2. Performance Parameter

In order to compare the various tritium detection techniques discussed below, it is necessary to understand the performance criteria for radiation detection. A brief explanation of these symbols is shown in Table 2.

The minimum detectable activity (MDA) is the lowest level of radioactivity that can be distinguished above background levels and within a given statistical 2σ confidence interval. The standard method for calculating the MDA was proposed by Currie in 1968 [12]. There is a 5% probability of the risk of non-detection and a 5% probability of the risk of false detection. “*T*_1_” represents the total counting time (in minutes), “*B*” represents the background count rate (in minutes), “*ε*” stands for the counting efficiency, expressed as a percentage, which is the ratio of detected radioactive particles (*q_D_*) to all the radioactive particles emitted from a source (*q_E_*), and “*Vs*” denotes the sample volume (in liters). The MDA is expressed in units of Bq/L.
(1)$$\mathrm{MDA}=\frac{2.71+3.29\sqrt{T_{1}\cdot B}}{60\cdot T_{1}\cdot \varepsilon \cdot V_{s}}$$
(2)$$\varepsilon =\frac{q_{D}}{q_{E}}$$

The separation coefficients *β_H/D_* and *β_H/T_* represent the degree of separation between hydrogen and its isotopes, deuterium (D) and tritium (T), during the gas–liquid phase. These coefficients also describe the enrichment effect of hydrogen and its isotopes. The expression for these coefficients is derived from the Arrhenius equation [13]. A larger separation coefficient indicates more effective separation.
(3)$$\begin{array}{c} \beta_{\frac{H}{D}}=exp\left(\frac{\Delta {ZPE}_{HD}}{k_{B}T}\right)\\ \; \end{array}$$
(4)$$\beta_{\frac{H}{T}}=exp\left(\frac{\Delta {ZPE}_{HT}}{k_{B}T_{0}}\right)$$
(5)$$\Delta {ZPE}_{HD}={ZPE}_{H}-{ZPE}_{D}$$
(6)$$\Delta {ZPE}_{HT}={ZPE}_{H}-{ZPE}_{T}$$

In the theoretical separation coefficient equations above, *ΔZPE _HD_* and *ΔZPE _HT_* represent the zero-point vibrational energy differences between hydrogen (H) and deuterium (D) and hydrogen (H) and tritium (T), respectively. Here, *k_B_* stands for the Boltzmann constant and T_0_ denotes the thermodynamic temperature.

## 3. Overview of Low-Concentration Tritium Technology

We counted the relevant studies on tritium in the aqueous environment within the last 5 years, and this section explores both pretreatment and detection. Overall, the amount of literature related to online measurement is low. More recent attention has focused on the provision of the detection side. In this section, the techniques are reviewed in terms of both preprocessing and detection. See Figure 2 for the bar chart.

### 3.1. Tritium Pretreatment Techniques

To overcome the detection challenges posed by low tritium concentrations, tritium pretreatment techniques have been developed. These methods are essential for enriching samples and minimizing interference from impurities, thereby facilitating more accurate subsequent measurements. Environmental water samples, in contrast to those from reactors, typically exhibit low tritium levels, a high presence of impurities, and a multitude of complex factors that influence measurements. Several enrichment methods are currently in use, including electrolysis, distillation, and the application of reverse osmosis membranes. However, distillation has been largely dismissed due to its requirement for bulky equipment and the extensive time it demands, which does not align with our research objectives. Solid polymer electrolyte (SPE) membranes have gained prominence in the realm of ultra-low background tritium monitoring, with ongoing research aimed at refining the materials and conditions for electrolysis to maximize the separation efficiency. The reverse osmosis membrane technique, although less documented and not yet widely adopted, shows promise as a novel pretreatment method. Its potential to significantly shorten sample processing times positions it as a candidate for carrying out the enrichment processes required for the advancement of future online monitoring technologies.

#### 3.1.1. Electrolytic Method

Electrolysis is currently the most mature method for enriching low concentrations of tritium. It is classified into alkaline electrolysis (AE), solid oxide electrolysis (SOE), and solid polymer electrolyte electrolysis (SPE), based on the electrolyte, operating conditions, and charge carriers (OH^−^, O^2−^, H^+^). Alkaline electrolysis is the earliest and most mature electrolysis technology, while the SPE method is more efficient and safer than the AE method. The current SOE technology is limited by the immaturity of the material properties [15]. The SOE method is limited by the immature nature of the material and is not considered here. On the other hand, the SPE method surpasses the AE method in terms of measurement efficiency and safety, making it the preferred choice for practical applications.
Alkaline Electrolysis (AE)

Alkaline electrolysis is considered to be one of the earliest and most developed electrolysis technologies [16,17]. It typically consists of a cathode, an anode, a diaphragm, and an electrolyte. A 20% to 30% solution of potassium hydroxide (KOH) is commonly used as the electrolyte, with asbestos serving as the diaphragm to aid in the production of H_2_ and O_2_. Transition group elements, like Ni and Fe, are often used as catalysts in alkaline electrolysis tanks due to their cost effectiveness and easy availability. The operating temperature usually ranges from 20 °C to 80 °C. The structure of alkaline electrolysis tanks is simple and cost effective, making it a technically mature method. However, despite its historical use in tritium concentration pretreatment, the alkaline electrolysis method is gradually being phased out due to its low efficiency, alkali treatment limitations, and gas deflagration issues.
2.Solid Polymer Electrolyte (SPE)

Solid polymer electrolyte (SPE) electrolysis was proposed by Grubb in the 1950s and developed by General Electric in 1966 [18]. SPE electrolysis utilizes a solid polymer electrolyte membrane to conduct protons, which is a proton exchange membrane (PEM), known as a perfluorosulfonate proton exchange membrane (PFPEM). Therefore, SPE electrolysis is also known as PEM electrolysis. The solid polymer electrolyte membrane effectively isolates gases on both sides of the electrode, eliminates the need for a liquid electrolyte, and avoids the drawbacks associated with the use of strong alkaline fluids. The SPE water electrolysis method holds significant application value due to its zero-pitch structure and lower ohmic resistance, which greatly enhances the overall efficiency of the electrolysis process, as well as its more compact size and lower operating temperature (20~80 °C), high safety, and easy operation. Currently, SPE electrolyzers mainly use precious metal catalysts, such as platinum and palladium, as hydrogen precipitation reaction materials and iridium and ruthenium as oxygen precipitation reaction materials. Therefore, synthesizing the available literature, reducing the production costs and improving the service life are the main challenges in terms of SPE electrolysis at present; also in 20~80 °C environmental conditions, electrolysis efficiency increased to 65~82%, due to the SPE membrane that separates the H_2_ and O_2_, which means that safety has been improved, thus the solid polymer electrolyte electrolysis can be considered as an advancement over of the alkaline electrolysis method. See Figure 3 for the schematic diagram.

Saito et al. (1996) were the first to use SPE membranes for experiments on the electrolytic concentration of tritium [19]. They successfully developed an automatic shutdown SPE tritium electrolytic concentration device. Suntarapai (1998) conducted electrolysis experiments using a constant current of 50 A. When 1000 mL of the initial volume of a water sample was electrolyzed to 75 mL, the separation factor β_H/T_ was 5.78 [20]. Ivanchuk et al. (2000) used platinum (Pt) as an electrode catalyst to investigate the relationship between temperature and the separation factor [21]. When the temperature range was 33 °C~At 298 K, the β_H/T_ values obtained were 4.95~6.9. Bellanger (2007) utilized Au/Pt membrane electrodes and adjusted the current density of the SPE electrolysis cell to 75 mA/cm^2^, leading to a β_H/T_ value of 6.1 [22]. Aurelie (2009) utilized seven series-connected SPE electrolysis cells with Ir/Pt membrane electrodes and achieved a β_H/T_ value of 4.6 ± 0.3 for the system, operating at a water temperature of 35 °C [23]. Up to this point in the development of the technology, the focus of the research has gradually shifted from the electrolysis conditions to the electrolysis materials, and research at the material level has improved the separation effect of H and T.

In recent years, Lozada-Hidalgo studied the incorporation of graphene and hexagonal boron nitride into membrane electrodes [24,25]. This study demonstrated improved hydrogen (H) and deuterium (D) separation performance, achieving a separation coefficient (*β_H/D_*) of 10. A tritium enrichment system in water, based on the combination of a sulfonated poly(ether ether ketone) (SPEEK) membrane electrolytic cell and a fuel cell, was developed by R.J. Zabolockis et al. After adding graphene to the membrane, the tritium separation factor *β_H/T_* increased from 10 ± 1 without graphene to 20 ± 4 with graphene [26]. It is expected that the separation factor will be further improved in the future with the development of chemical membrane material research.

#### 3.1.2. Reverse Osmosis (RO) Film

The reverse osmosis membrane method is most commonly used for the desalination of seawater [27,28]. This method was first used for LSC preprocessing measurements in 2004, by Takayuki Koganezawa et al. [29]. The use of reverse osmosis membranes as a pretreatment method for LSC was shown to be a feasible approach, based on an experimental study. The study indicated that this method could decrease the pretreatment duration from 2 days to 1.5 h. Azevedo et al. studied a tritium real-time monitoring system deployed at the Almaraz Nuclear Power Plant (ANPP) in Spain, from 2018 to 2020. They employed the addition of NaClO and activated carbon to treat ambient water during tritium sample enrichment, using reverse osmosis membranes. Additionally, they used UV sterilization for further treatment of the enriched tritium. The study reported that there was no loss of beta radioactivity when measuring tritium through these water treatment processes. The report emphasizes that there is no loss of beta radioactivity in terms of the tritium measurements with such water treatment processes. 

Li et al. at Tsinghua University, in 2020, studied and validated a reverse osmosis (RO) membrane as a means of LSC pretreatment, verifying and optimizing the performance parameters of the RO membrane pretreatment unit [30]. This report is one of the few research efforts on preprocessed tritium nuclides. The report indicated that the service life of the RO membrane exceeded 70 cycles, the sample consumption was approximately 13 L, and the average treatment time was 40 min. This resulted in a reduction in the pretreatment time by over 77% compared to the distillation pretreatment method. Tritium residue on the reverse osmosis (RO) membrane can be ignored when treating environmental water samples using this method. However, it is not advisable to use this method for pretreating water samples with a high tritium concentration (>10 Bq/L).

Research on the use of reverse osmosis (RO) membranes for pretreatment has shown that in addition to the existing papers mentioned above, there may be a number of potential issues that need to be brought to attention in the context of subsequent research: 1. the physical strength of the RO membranes and whether they retain the target substances, 2. the applicability of lower radioactivity in regard to the samples, and 3. the range of indicators or optimal indicator conditions in terms of the requirements of different RO membranes in the aqueous environment. The comparison of methods is shown in Table 3.

### 3.2. Tritium Detection Techniques

The field of tritium detection technology is tasked with accurately measuring enriched samples. In the spectrum of existing methodologies, liquid scintillation counting (LSC) stands out for its superior detection efficiency and low minimum detectable activity (MDA). Nonetheless, the LSC method is not without its drawbacks, notably the generation of radioactive organic waste. To circumvent this issue, the adoption of solid scintillators has emerged as a viable alternative, complemented by strategies for material recycling that support the feasibility of online monitoring systems. Crafting a detection apparatus that meticulously accounts for the scintillator’s material composition and physical attributes, while also streamlining the detection protocol to boost efficiency and minimize the MDA, represents a formidable technical hurdle in the advancement of real-time monitoring technologies.

#### 3.2.1. Liquid Scintillation Counting (LSC)

Liquid scintillation counting is a commonly used method for measuring tritium in water and has become an internationally recognized standard [32]. The scintillation fluid consists of a solvent, a first scintillator, and a second scintillator. Usually, the scintillation solution is mixed with pretreated tritium samples. The β-rays generated by tritium decay are excited by the scintillation solution solvent. The energy emitted is absorbed by the solute and then transformed into photons after the excitation and de-excitation process. The tritium concentration and radioactivity level can be measured by the photomultiplier tube (PMT) in the liquid scintillation analyzer through electrical signal processing. See Figure 4 for the schematic diagram.

The method is technically mature, offering high measurement efficiency and a low detection limit, making it suitable for measuring low concentrations of beta nuclides. It is usually used to measure the radioactivity level of β-nuclides in low concentrations. Over the past 60–70 years [32,33,34], many studies using practical LSC methods for measuring ^3^H sources have been carried out and, until within the last 10 years, research has been centered around lowering the lower detection limit of the LSC method [35,36,37], and based on this paper’s discussion on measuring low-activity ^3^H in water, the scope is limited to a summary of the explorations made within the last 5–10 years on the improvement and optimization of the LSC method.

Current research based on the LSC method mainly focuses on optimizing the LSC measurement strategy. Some commercially available instruments for measuring low tritium concentrations already have a good figure of merit (FOM [38]), as well as a low MDA. In recent years, researchers have developed various measurement schemes for the LSC method. These schemes involve adjusting the parameters, such as the pretreatment, sample volume, and sample-to-scintillant volume ratio, etc., based on the specific characteristics of the samples under study. The goal is to enhance the accuracy of the measurement results obtained through the LSC method and to minimize the MDA.
(7)$$FOM=\frac{\varepsilon^{2}}{B}$$

The ε represents the detection efficiency (count rate/decay rate) and *B* stands for the background count rate (cpm).

The following are typical studies on improving LSC performance in recent years:

Lin et al. (2020) developed a tritium measurement method in seawater, using SPE technology for electrolytic enrichment and ultra-low background liquid scintillation counting (LSC), with a Quantulus 1220 spectrometer from PerkinElmer [39]. They conducted measurements on seawater samples collected from the Arctic Ocean. The study suggested that electrolytic treatment of 350 mL of the sample with a sample-to-scintillation liquid ratio of 8:12 offers the best detection efficiency and the method has a low MDA of 0.10 Bq/L.

Based on the AccuFLEX LSC-LB7 measurement system and using 100 mL liquid scintillation counting vials, Feng et al. explored the optimal measurement conditions and strategies for tritium in the measurement environment [40]. They investigated the optimal dark adaptation time of the instrument, the mixing ratio of the scintillation solution and the sample volume, and provided recommendations for the measurements: a sample volume below 15 mL is not recommended for measurement and specific activity below 0.4 Bq/L is advised for enrichment treatment before measurement.

In 2022, there are reports on gaseous tritium measurements in online monitoring [41]. However, an online monitoring device for a low tritium concentration in water based on the LSC method has not been reported in the literature, despite the proposal on a computerized, automated control scheme by Sigg in 1994 [42], and the commercially available online monitoring device LabLogic Wilma On-line LSC. The introduction of such an instrument indicates that the LSC-based online measurement technology has become more mature and more related measurement instruments will appear from the vision of researchers in the future. See Figure 5 for the relevant product diagram.

#### 3.2.2. Plastic Scintillators (PSs)

A plastic scintillator is a polymer-based material that can have its properties modified by the intentional addition of new substances. It is one of the most promising and versatile radiometric materials in use today. It offers a fast response, high detection efficiency, and low manufacturing and instrumentation costs. Common plastic scintillators and their parameters can be seen in Table 4. Additionally, it provides a reusable alternative to other detection materials [38]. Like inorganic crystalline scintillators, it has the potential to serve as a replacement for LSC. A flake, fiber, or microsphere scintillator morphology offers the advantage of a large specific surface area and specific volume. Therefore, common types include flake plastic scintillators, plastic scintillator fibers, and microsphere plastic scintillators. Plastic scintillators have more quenching factors than LSCs, such as distance quenching. For the measurement of low-energy β-nuclides, the β-rays generated are attenuated before reaching the surface of the scintillator, resulting in decay, which leads to very low detection efficiency. In 1995, Hofstetter reported on a mobile radiation detection system based on plastic scintillator microspheres, but the detection efficiency was only 0.3% [43].

We have selected some commonly used plastic scintillator materials as alternatives for making detectors, and a comparison of their performance is provided in Table 5.

The relevant literature shows that plastic scintillators have the advantage of high optical yield compared to LSCs, but they also have an obvious drawback: the detection efficiency of solid scintillators is significantly lower than that of liquid scintillators. This is mainly related to the relative contact area of the solid scintillator with the measurement sample, so in order to increase the contact area, solid scintillator particles, plate solid scintillators, and other forms of application have appeared [47]. In recent years, there have been a small number of studies on synthesizing and improving fluorescent substances in plastic scintillators, with the aim of improving the scintillator performance mainly at the material level, but if the relative contact area between the sample and the scintillator is neglected, even with high luminescence efficiencies, the detection efficiency will be low and the final measurements will be less than satisfactory [48,49]. 

Etsuko Furuta et al. conducted research on plastic scintillators [50]. In 2014, they introduced a sheet plastic scintillator treated with a plasma surface treatment. The plastic scintillator used in the experiment is shown in Figure 6. The analysis involved holding a small sample amount (5 μL) between two sheets and placing it into a low-background liquid scintillation analyzer assay. The detection efficiency was found to be 35%. In 2017, they reported on a study that used a spherical plastic scintillator instead of a liquid flash reagent [51,52]. Comparing it with the previously used plasma-treated plastic scintillator, they found that the detection limit and counting efficiency of the spherical plastic scintillator (45%) were higher than that of the flake scintillator. However, an increase in the amount of water used would decrease the detection efficiency. To achieve the maximum detection efficiency, a longer measurement time would be required, specifically for the lower detection limit of 5 Bq/mL at 10 min, the lower detection limit was 3.2 Bq/mL at 100 min.

Azevedo et al. designed a detector based on plastic scintillation fibers (polystyrene) [53,54]. The detector has a total of five cells, with each cell containing approximately 340 scintillation fibers, measuring 2 mm in diameter and 18 cm in length. The entire measurement system operates in real-time, with a measurement period of 1 h, and its MDA can reach 100 Bq/L. The materials used in the research project are shown in Figure 7.

A feasibility study on the plastic scintillator particles, involving static measurements and mobility experiments, was conducted by Jiri Janda et al. in 2022 in the Czech Republic [55]. The study revealed that when detecting low-energy β particles, the size of the microspheres decreases, leading to a reduction in the burst and an enhancement in the detection efficiency. Due to the low β-emission energy of tritium, the maximum detection efficiency for tritium was 3.9 ± 0.2%, which was significantly lower than the maximum 90 Sr/Y detection efficiency of 71.2 ± 3.9%.

In 2023, the Fukushima Daiichi nuclear power plant in Japan utilized plastic scintillators for fundamental research on tritium in treated water discharge [56]. The study employed both sheet and granular scintillators (Figure 8). The depth of the β-ray penetration in plastic scintillators was estimated to be up to 10 μm, through rough analysis using the Monte Carlo method. It was observed that the greater the area of contact between the PS and the liquid, the higher the efficiency. The experiments were conducted over a period of 3600 s, and the MDA of the granular PS was determined to be 3240 Bq/L, while for the lamellar PS it was 7500 Bq/L. Furthermore, it was noted that the MDA decreased with longer measurement times.

Yukihisa Sanada et al. [57] developed a practical tritium monitor for the continuous monitoring of the tritium concentration in treated discharge water from the Fukushima Daiichi nuclear power plant (Figure 9A). The monitor contains a flow cell detector comprising plastic scintillator particles, along with simultaneous measurements from three detectors: an anti-compliance detector and lead shielding. The system achieved an MDA of 593 Bq/L in 60 min.

#### 3.2.3. Inorganic Scintillators (CaF_2_:Eu)

The greatest advantage of the inorganic solid material, CaF_2_:Eu, is that it does not react with water, is not susceptible to deliquescence, and has a low atomic coefficient [58]. These properties make it an ideal inorganic material for measuring low-energy β in liquids and a potential alternative to the LSC method for reducing radioactive waste liquids. The CaF_2_:Eu morphologies reported in the literature, which have been effectively measured so far, are in the form of powder granules [59] and flakes. See Figure 10.

A. N. Vasil’ev first studied and discussed the factors determining the efficiency of inorganic scintillators in 2014 [60], analyzing the relationship between the scintillator size and luminescence efficiency at a theoretical level. Takao Kawano et al. created three measuring tubes connected in parallel with the CaF_2_:Eu, filled with CaF_2_:Eu powder [61]. They allowed the sample to flow through the tubes to complete the luminescence. The experiment explored the effect of different flow rates on the counting. The literature reported that a minimum of 10 Bq/mL could be detected with 1000 s of measurement time, and the change associated with time could be expressed linearly.

In recent years, inorganic scintillators have been studied from the following perspectives: Designing the optimal shape of scintillators for their application in conjunction with Monte Carlo simulation;Building and designing detectors to test the detection capability;Improving existing detectors, so as to reduce the MDA.

Alton, Tilly Lucy, from Lancaster University, UK, explored the luminescence efficiency of grinding CaF_2_:Eu into particles of various sizes using different methods [62]. The powder was used to detect tritium in groundwater and, in combination with GEANT4 simulations, determined the particle size corresponding to tritium with a maximum energy deposition of 3.5 μm. In 2019, Jing Wu from the University of Technology, Chinese Academy of Sciences, enhanced this method by using CaF_2_:Eu crystal powder, constructing a photonic pathway with a wave-shifting optical fiber and creating a prototype [63,64]. The research in the literature indicates a tritium–water volume fractions of 61.89% and 74.45% for the 8 μm diameter powder utilized, with a measurement duration of 10 min and a background count of 2400. The method achieved an MDA as low as 0.64 Bq/mL. 

Song et al., at the University of Science and Technology of China (USTC), in 2021, investigated counting chambers comprising sheets of CaF_2_:Eu [65]. Each basic unit of the detector consists of a pair of CaF_2_:Eu scintillator sheets and a sample chamber for holding tritium water, see Figure 11. The outer layer of the detector was wrapped with aluminum foil to prevent light interference and reflect scintillation photons. These photons were detected by photomultiplier tubes at both ends. The study resulted in an MDA of 2.95 Bq/mL for 60 min for one design option, as determined through simulation.

The above studies on plastic scintillators and inorganic scintillators (CaF_2_:Eu) have shown increasing detection efficiency and sensitivity, with some designs featuring real-time measurements. These scintillators are available in a variety of shapes, such as flakes, spheres, and pellets, expanding their suitability for specific applications. However, the detection efficiency of plastic scintillators may be compromised by the presence of moisture. In addition, the detection efficiency of tritium is still low compared to other radionuclides, such as 90 Sr/Y. Longer measurement times are necessary to achieve lower minimum detectable activity (MDA) values. To address these challenges, future research should focus on optimizing scintillator materials and geometries to improve detection efficiency, especially for tritium. The development of more efficient waterproof coatings or treatments could counteract the negative effect of water on efficiency. The use of advanced data processing techniques could improve real-time analysis and reduce the required measurement time. In addition, research into the synergistic use of different types of scintillators for multi-isotope detection could improve the sensitivity to a wider range of radionuclides.

## 4. Discussion

This review systematically examines the preprocessing and detection methodologies, with a focus on their suitability for online tritium measurement. The overarching aim of these preprocessing techniques is to enhance the fidelity of subsequent detection outcomes. Our literature synthesis has underscored the significance of measurement and cycle times as critical performance indicators. A comparative analysis reveals that electrolysis, and notably the solid polymer electrolyte (SPE) method, predominates in studies dedicated to detection techniques, closely followed by distillation as a streamlined pretreatment option for online monitoring. Electrolysis offers a more compact setup and demonstrates superior consistency and efficiency in terms of processing times over distillation methods. While electrolytic techniques have demonstrated robust enrichment capabilities, the scope for further technological refinement appears to be reaching its limit. To advance enrichment separation factors and accelerate processing speeds, innovative approaches involving catalyst materials, advanced membrane electrodes, and novel two-dimensional materials, such as graphene (Gra) and hexagonal boron nitride (h-BN), are anticipated to drive theoretical and practical breakthroughs.

The incorporation of reverse osmosis membranes represents a novel frontier in pretreatment strategies. Given the distinct nature of this technology compared to those previously discussed, direct comparisons with established online monitoring pretreatment methods are challenging. The method’s primary allure lies in its enhanced processing velocity, yet its full potential remains to be delineated and rigorously assessed. Reverse osmosis membranes are recognized for their user friendliness and automation in seawater desalination and their application for radiation measurement pretreatment holds promise for mitigating lengthy processing times. However, preliminary evaluations indicate stringent water sample prerequisites for this method: an initial conductivity ranging from 6.98 to 919 μS/cm and turbidity levels below 1.0 NTU. These specifications may be demanding for certain water samples, necessitating additional research and validation to ascertain the method’s pretreatment efficacy and optimal operational parameters. Relevant technical parameter indicators are presented in Table 3.

The detection techniques mentioned, encompassing both liquid and solid scintillator counting, are notable for their effectiveness. However, liquid scintillation counting (LSC) stands out as the most significant method, as evidenced by comparative analysis of the performance results from the literature and experimental data. With nearly seven decades of development, LSC technology has reached a plateau in the measurement of low-energy nuclides. This maturity is attributed to robust technical and theoretical foundations, coupled with superior measurement capabilities. However, researchers should consider several significant challenges associated with the use of liquid scintillation counting (LSC) or solid-state scintillator technology for online monitoring in their subsequent work.
The time measurement gap in systematic MDA optimization studies

For different systems and measurement conditions there exists a locally optimal policy: a locally optimal solution that guarantees the lowest MDA in the smallest possible measurement time. There is currently little research on such issues. Studies by Erchinger et al., Lin et al., and Sigg et al. have consistently demonstrated an inverse relationship between the measurement time and the lower minimum detectable activity (MDA). Prolonging the measurement period to enhance performance inherently extends the timeframe, potentially compromising the expected MDA if measurements are conducted over shorter durations. The meticulous sample pretreatment process itself requires a minimum of 3–5 h to achieve a lower MDA, a timeframe that may not align with the exigencies of emergency monitoring. Despite advancements, such as the LabLogic Wilma system for tritium water online monitoring, there is scant literature and experimental data to validate the performance of these systems, underscoring the need for further research to refine the balance between measurement time and MDA, tailored to diverse application scenarios.
2.Monitoring system of organic radioactive waste generation is easily overlooked

The reduction of radioactive waste generation during sample preparation is an unavoidable problem associated with the LSC approach. The generation of organic radioactive waste during LSC measurement is an unavoidable byproduct, primarily stemming from liquid flash and radioactive samples. Estimates from the literature suggest that the waste volume from a single batch of measurements is typically under 500 mL. For instance, Feng et al. estimated less than 200 mL of waste liquid per measurement, excluding parallel control groups. Similarly, C. Varlam et al. [66]. reported an estimate of less than 250 mL per measurement. Scaling this to a 5 h online measurement cycle with five parallel groups, the projected weekly waste liquid volume reaches approximately 85 L. While manageable for short-term applications, the annual waste volume for water environment monitoring around nuclear power plants could exceed 4320 L, not accounting for additional waste from instrument cleaning. This scale of waste generation contradicts the environmental protection ethos and poses a significant concern. A strategic shift in the measurement approach is imperative. The literature also highlights the potential of solid scintillators as a promising, waste-free alternative to LSC technology.

The literature on solid scintillators, as previously discussed, highlights their common advantages of reusability and minimal waste production. Various forms, such as plates, powder particles, and scintillation fibers, have been explored in studies. Among these, powder particles stand out due to their high geometrical efficiency when in contact with tritium solutions. However, a side-by-side comparison indicates that the detection capabilities of solid scintillators are not yet on a par with those of liquid scintillation counting (LSC). Furthermore, there are several issues with the studies concerning the measurement of tritium using plastic and inorganic solid scintillators.

Compared to LSC-related research, there is a need for a more extensive accumulation of the literature on this subject over time. These technologies have not yet been integrated into commercial-grade products and practical systems, with the exception of the measurement system based on plastic scintillation fibers (polystyrene) developed by Azevedo et al. Additionally, some of the experimental effects reported in the literature are inefficient and overly idealized.
3.A failure to distinguish non-tritiated nuclide interferences in environmental samples and fast solutions for machine learning

The complexity of environmental water samples may have been underappreciated in the literature, particularly in studies that rely on specially configured specimen samples. This oversight is primarily due to a lack of focus on beta radionuclides other than tritium. In the studies by Akata, N [67]. et al., Zhang et al. [68], and Lin et al., the reported results for the environmental samples may reflect the total beta radioactivity rather than the tritium activity alone. When employing environmental samples for measurement, it is imperative to consider the following points more thoroughly:(1)Tritium is not the sole radioactive element present. The contribution of other low-energy beta-emitting isotopes, such as ^14^C, to the total beta measurement results must be ascertained;(2)Interference from alpha-decaying nuclides, such as radon, is a real concern. The extent to which these nuclides affect the measurement outcomes remains to be determined;(3)The impact of background radiation on the measurement environment is also a significant factor that requires careful assessment.

To address the aforementioned challenges, mitigating background radiation interference is the most straightforward task and can be achieved by implementing shielding measures. However, dealing with interference from non-tritium nuclides requires a more nuanced approach, such as incorporating energy spectrum analysis for nuclide identification. This could potentially increase the cost and time associated with measurements.

Looking ahead, the integration of artificial intelligence (AI) and neural networks into nuclide identification studies may offer a promising solution for accelerating the measurement process. There is already evidence of progress in this area: an algorithm based on artificial neural networks (ANNs) [69,70] has been developed and is reported to rapidly predict the activity ratios of ^3^H and ^14^C, with an accuracy that deviates by approximately 5%.

## 5. Conclusions

The objective of this review is to delve deeply into the preprocessing and measurement technologies required for online monitoring, identifying the current challenges and charting potential future trajectories for the advancement of online monitoring instrumentation for environmental water samples. Initially, by meticulously reviewing and analyzing a substantial body of recent literature, we dissect the preprocessing and detection methodologies, weigh their respective merits and demerits, and assess their potential capabilities and prospective development avenues within the realm of online monitoring.

At present, LSC detection technology with SPE pretreatment is the most applicable solution in the field of online monitoring, as it is the only commercialization of the technology program, other technologies, such as plastic scintillators and inorganic scintillators, need to be further prototype tested to achieve online monitoring applications or commercialization. Future research in further reducing the lower limit of detection and detection efficiency of the technology related to the following points is needed: 1. An online monitoring system optimization strategy, based on LSC technology to balance the measurement time and MDA, so that the system better fits the actual conditions; 2. plastic scintillator and inorganic scintillator methods, as an alternative to the LSC measurement method to reduce organic waste generation; 3. The combination of machine learning to achieve the rapid screening of interfering nuclides and to improve the speed of analysis. 

In conclusion, this review underscores the significance of tritium in the environment, its importance in water environmental monitoring, and the critical importance of research into pertinent online monitoring techniques. It highlights the current research milestones and the challenges faced in this domain. As technology continues to evolve, further in-depth exploration of these issues will establish a robust foundation for the deployment of online monitoring systems.

## Figures and Tables

**Figure 1 sensors-24-05722-f001:**
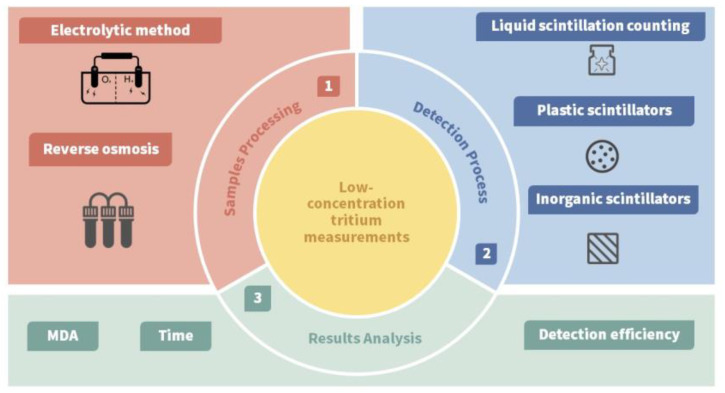
Flowchart for low-concentration tritium measurement and review abstract.

**Figure 2 sensors-24-05722-f002:**
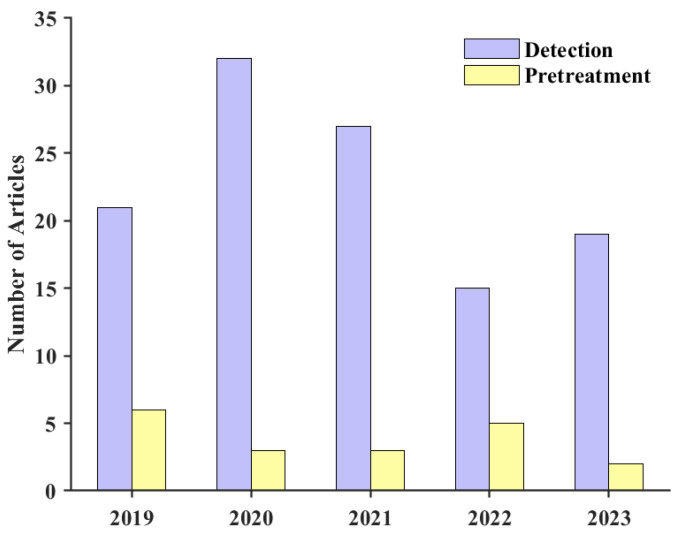
Statistics on the amount of research on tritium-related studies.

**Figure 3 sensors-24-05722-f003:**
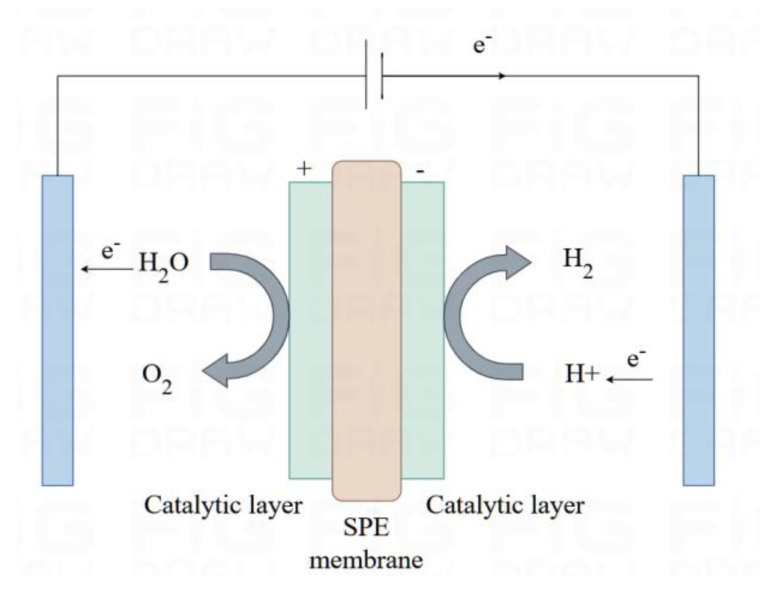
Diagram of SPE membrane electrode working principle.

**Figure 4 sensors-24-05722-f004:**
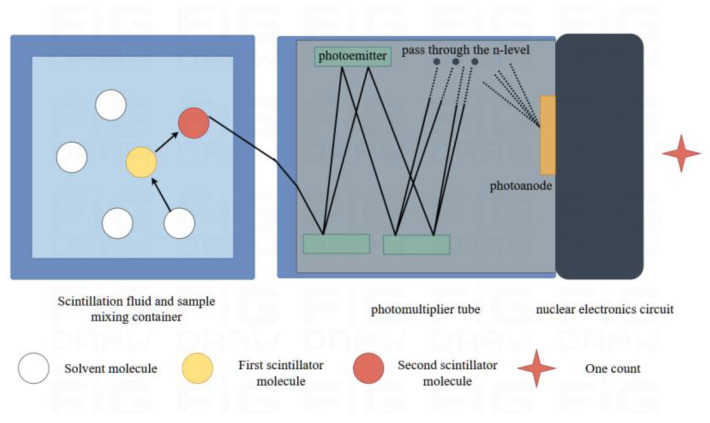
Liquid scintillation counting schematic.

**Figure 5 sensors-24-05722-f005:**
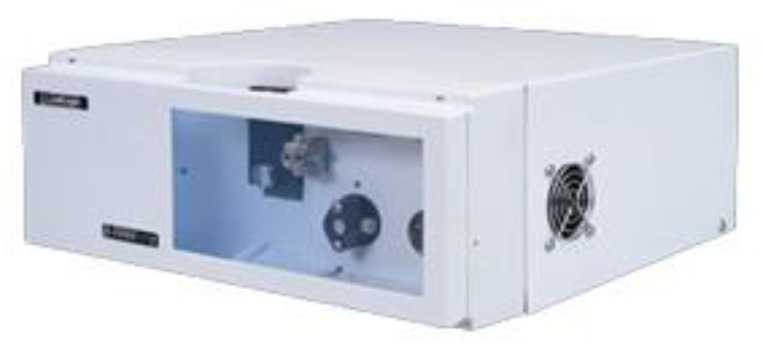
LabLogic Wilma On-line LSC.

**Figure 6 sensors-24-05722-f006:**
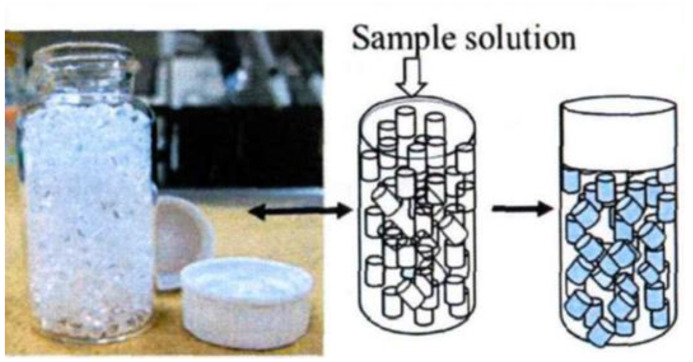
Plastic scintillator spheres for measuring tritiated water, by Etsuko Furuta et al.

**Figure 7 sensors-24-05722-f007:**
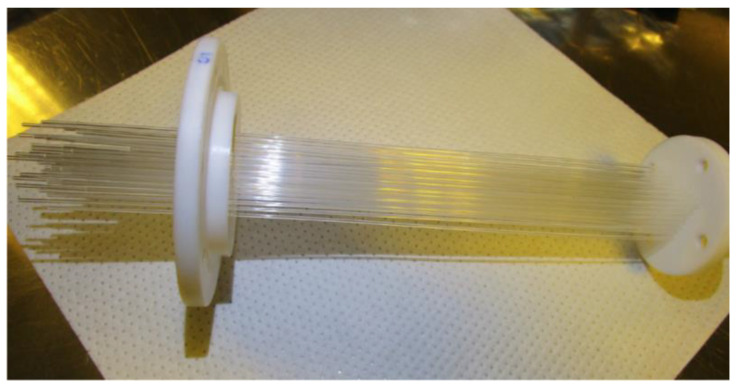
Scintillating fibers used in measurement system by Azevedo et al.

**Figure 8 sensors-24-05722-f008:**
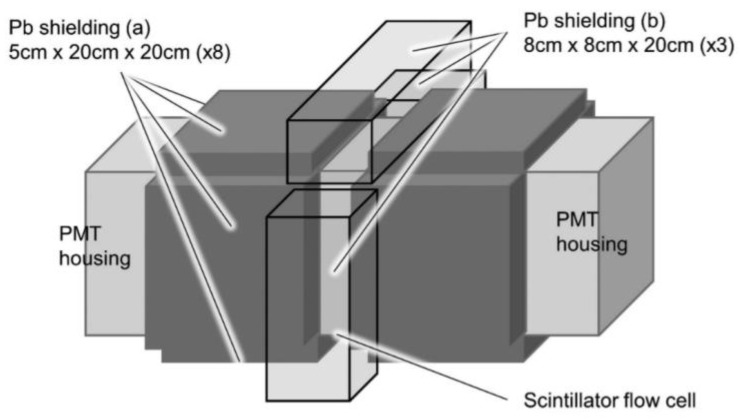
A diagram and shielding design of the device by Fukushima Daiichi et al.

**Figure 9 sensors-24-05722-f009:**
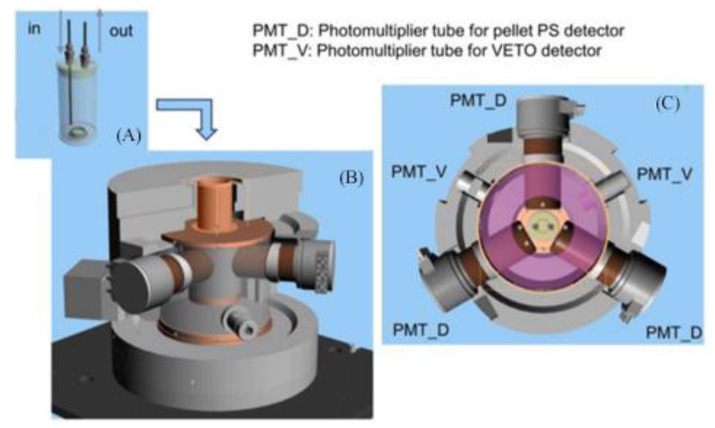
(**A–C**) Design of a plastic scintillator-based flow cell detection device by Yukihisa Sanada et al.

**Figure 10 sensors-24-05722-f010:**
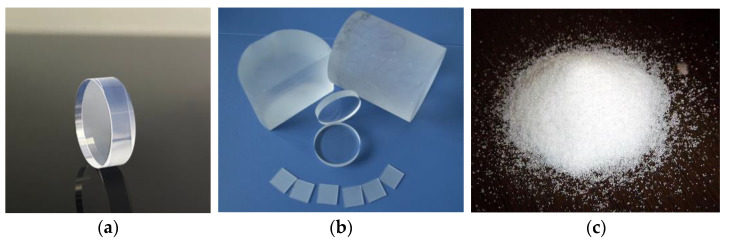
CaF_2_:Eu crystals in different application shapes: (**a**) CaF_2_:Eu crystal for conventional detectors; (**b**) plate, block, sheet CaF_2_:Eu crystals; (**c**) powdery CaF_2_:Eu crystals.

**Figure 11 sensors-24-05722-f011:**
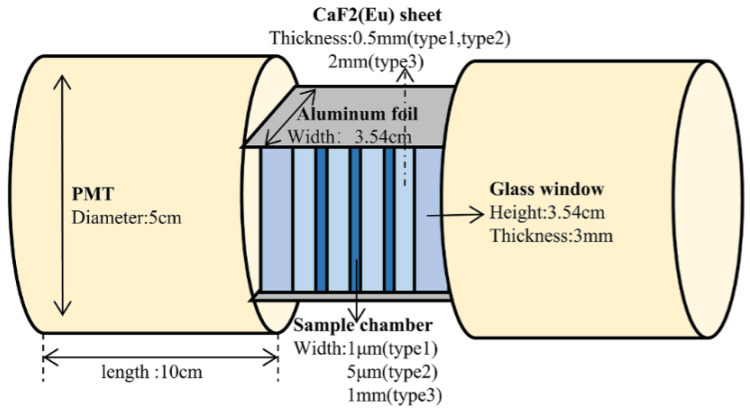
Chamber in Monte Carlo simulation by Song et al.

**Table 2 sensors-24-05722-t002:** Overview of the features.

Categorization	Type of Technology	Key Features
Pretreatment	Electrolytic method	Common method, easy to setup
	Reverse osmosis (RO) film	Newly proposed method in recent years, with shorter processing time than electrolysis
Detection	Liquid scintillation counting (LSC)	The most mature and best performing method for measuring low-energy radiation signals
	Plastic scintillators (PSs)	Premature, no organic waste compared to LSC
	Inorganic scintillators (CaF_2_:Eu)	Premature, no organic waste compared to LSC

**Table 3 sensors-24-05722-t003:** Performance comparison of the different pretreatment technologies.

Pretreatment Technical Method	Effect	Estimated Processing Speed
Alkaline electrolysis	Optimal *β_H/T_* about 10, electrolytic efficiency 59~70% [31]	-
Solid polymer electrolyte	Optimal *β_H/T_* about 20, electrolytic efficiency 65~82% [14]	14 mL/h
Reverse osmosis (RO) film	Can handle water samples below 10 Bq/L	36 mL/min

**Table 4 sensors-24-05722-t004:** Commonly used plastic scintillator materials [44,45].

Material	BC408	PMMA	PS (Polystyrene)	VMB (Vinyl Methyl Benzene)	LiME (Lithium Methacrylate)	EJ-309 (A Liquid Scintillator Used in LSC)
Relative light yield (100%)	116.8079	85.20481	115.179	111.2911	108.153	90.813
Detection efficiency	0.41%	0.41%	0.41%	0.40%	0.40%	95.9%

**Table 5 sensors-24-05722-t005:** Performance comparison of different detection technologies.

Technical Detection Method	Estimated MDA	Detection Time	ε	MDA < EPA (740 Bq/L)	MDA < EC (100 Bq/L)
Liquid scintillation counting (LSC)	0.6 Bq/L [46]	3 h	102%	√	√
Plastic scintillator powder (flow cell with Pb shielding)	593 Bq/L	1 h	5%	√	-
Plastic scintillation fibers	100 Bq/L	1 h	5%	√	-
CaF_2_:Eu powder (flow cell)	640 Bq/L	1 h	3%	√	-
CaF_2_:Eu sheets	2.95 Bq/mL	1 h	25%	-	-

## Data Availability

The data presented in this study are available in this article.

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
