# Peer review of "Measurement Techniques for Low-Concentration Tritium Radiation in Water: Review and Prospects"

_sensors, 2024, doi:10.3390/s24175722_

Round 1
Reviewer 1 Report
Comments and Suggestions for Authors
Dear Junxiang et al.:
Comment #01: None of the figures are in the text, this is a major flaw.
Comment #02: The introduction is very poor. I do not see clearly which are the most promising techniques nor a clear development of the review.
Comment #03: Improve the wording of the introduction, making the methods of titrio production that have been used clearer.
Comment #04: The introduction contains older and unnecessary references and should be limited to methodologies published in the last 5 to 10 years at most.
Comment #05: You should make a bar graph with the publications over the years, and how the number increases.
Comment #06: It is not clear to me what the motivation for the review is.
Comment #07: According to the review, what are the future prospects?
Author Response
Comment 1: None of the figures are in the text, this is a major flaw.
Response 1:Dear Reviewer, thank you for pointing out the absence of figures in the text, which is indeed a significant oversight. We have taken note of this and will include the necessary figures in the revised manuscript to support our arguments and data. We will ensure that all key data are presented graphically for a more intuitive understanding by the readers.
Comment 2:The introduction is very poor. I do not see clearly which are the most promising techniques nor a clear development of the review.
Response 2:We recognize that the introduction needs to more clearly present the prospects and methods of the research. In the revised manuscript, we will rewrite the introduction to more clearly outline the most promising techniques and provide a clearer graphical abstract for the development of the review.
Comment 3:Improve the wording of the introduction, making the methods of titrio production that have been used clearer.
Response 3:We recognized the introduction. In the revised manuscript, we reorganized the language of the introduction, added Table 3 and graphical abstract to highlight the article's point of view.
Comment 4: The introduction contains older and unnecessary references and should be limited to methodologies published in the last 5 to 10 years at most.
Response 4: We appreciate your comment on the inclusion of older references. Recognizing their value in establishing historical context, we have retained essential older citations that provide foundational support for our study. At the same time, we have pruned less relevant older references to emphasize the most recent methodologies.
Comment 5: You should make a bar graph with the publications over the years, and how the number increases.
Response 5: We have included a bar chart of the last 5 years of relevant research in the manuscript to demonstrate this area of research.
Comment 6: It is not clear to me what the motivation for the review is.
Response 6: We have reorganized the language of the introductory section to emphasize the context and the need for research.
Comment 7: According to the review, what are the future prospects?
Response 7: With regard to future prospects, we have made changes in the discussion and conclusion sections. In the first version of the manuscript, the language of the discussion focused too much on the problems and weakened the view of future development; in response, we have revised the discussion to point out the problems of current research, and in the conclusion, we have clarified the future research directions and potential development prospects of this review.
Reviewer 2 Report
Comments and Suggestions for Authors
This paper provides a comprehensive investigation for tritium sample preparation and detection techniques. The manuscript is well written, and presents an interesting scientific content. The paper can be accepted for publication in Sensors.
(1) What are the state-of-the-art challenges of the development of tritium sample detection, please also added author perspective to overcome those challenges.
(2) The authors are suggested to add some tables to compare the tritium sample detection techniques in your review manuscript.
Comments on the Quality of English LanguageThe manuscript is well written, and presents an interesting scientific content.
Author Response
Comment 1: What are the state-of-the-art challenges of the development of tritium sample detection, please also added author perspective to overcome those challenges.
Response 1: Dear Reviewer, thank you for your request to address the state-of-the-art
challenges in the development of tritium sample detection. We acknowledge the importance of highlighting these challenges and providing our perspective on overcoming them. In the revised manuscript, We have reorganized that outlines the current challenges:(1) MDA and time limitations;(2) Overlook of organic waste, (3) The need for need for nuclide identification. We also offer our insights into potential solutions, including the Balance measurement time and optimal MDA study. Promoting the study of plastic scintillator and inorganic scintillator systems as alternatives to organic waste free measurement methods, and Rapid nuclide identification using machine learning.
Comment 2: The authors are suggested to add some tables to compare the tritium sample detection techniques in your review manuscript.
Response 2: We have included a bar chart and Table 3 at the beginning of Chapter 3 to provide a brief summary of the literature and the characteristics of the relevant technologies
Reviewer 3 Report
Comments and Suggestions for Authors
1. The review needs to be reorganized. The introduction part should not have figures. The authors can consider just one graphical abstract at the end of the introduction.
2. All new references should be included.
3. At least one comparative table is required to have a view about the features of all recent research.
4. The permissions of images should be obtained.
5. Figure 3 is so weak and not sufficient.
6. Figure 11 is so weak and not sufficient.
7. The last table doesn’t have any caption.
8. Conclusion is short and doesn’t reflect any perspective.
Comments on the Quality of English Language1. The review needs to be reorganized. The introduction part should not have figures. The authors can consider just one graphical abstract at the end of the introduction.
2. All new references should be included.
3. At least one comparative table is required to have a view about the features of all recent research.
4. The permissions of images should be obtained.
5. Figure 3 is so weak and not sufficient.
6. Figure 11 is so weak and not sufficient.
7. The last table doesn’t have any caption.
8. Conclusion is short and doesn’t reflect any perspective.
Author Response
Comment 1: The review needs to be reorganized. The introduction part should not have figures. The authors can consider just one graphical abstract at the end of the introduction.
Response 1: Dear Reviewer, thank you for your guidance on the manuscript organization. We have restructured the review accordingly, removing one figure from the introduction and placing a single graphical abstract at the end of this section, in order to give readers a clearer understanding of the migration of Tritium in nature, the figure "Tritium transport in the natural environment" has been retained. This change aims to provide a clearer narrative flow and a concise summary of our work.
Comment 2: All new references should be included.
Response 2: We appreciate your comment on the inclusion of older references. Recognizing their value in establishing historical context, we have retained essential older citations that provide foundational support for our study. At the same time, we have pruned less relevant older references to emphasize the most recent methodologies.
Comment 3: At least one comparative table is required to have a view about the features of all recent research.
Response 3: We have included a bar chart and Table 3 at the beginning of Chapter 3 to provide a brief summary of the literature and the characteristics of the relevant technologies
Comment 4: The permissions of images should be obtained.
Response 4: Thank you for your attention to the details regarding image permissions. We understand the importance of adhering to copyright regulations and ensuring all images used in our manuscript are properly credited. We have taken immediate action and have now obtained the necessary permissions for all images included in the manuscript. We appreciate your guidance in maintaining the integrity of our publication.
Comment 5: Figure 3 is so weak and not sufficient.
Response 5: We deleted Figure 3 and redrew a graphical abstract.
Comment 6: Figure 11 is so weak and not sufficient.
Response 6: We have added images of other applied forms of CaF2:Eu to emphasize the plasticity and application of CaF2:Eu crystals
Comment 7: The last table doesn’t have any caption.
Response 7: We apologize for the oversight with the last table lacking a caption. We have now added a comprehensive caption to the table that clearly explains the data presented and its relevance to the study. Thank you for your valuable feedback that has helped us to improve the manuscript.
Comment 8: Conclusion is short and doesn’t reflect any perspective.
Response 8: Thank you for your feedback on the conclusion. We have now expanded the section to offer a more in-depth analysis and a forward-looking perspective, including potential future research directions and the significance of our findings within the field. This revised conclusion better reflects the scope and impact of our study.
Reviewer 4 Report
Comments and Suggestions for Authors
The review paper provides a comprehensive review of existing methods for the measurement of low-concentration tritium in water. However, the originality of the research could be questioned as the manuscript largely summarizes previous work without presenting novel findings or approaches. While the review covers a broad range of techniques, it lacks in-depth analysis of the recent advancements in tritium detection technology. For example, the discussion on solid-state scintillators could benefit from a more detailed examination of the latest materials and their comparative performance with traditional LSC methods. Besides, the manuscript would benefit from the inclusion of more graphical data, such as charts or graphs, to illustrate the performance parameters of different techniques. This would make the comparison of MDA, counting efficiency, and other metrics more accessible to readers. Moreover, the review should provide a more critical analysis of the limitations and potential pitfalls of each technique discussed. For instance, while the authors mention the safety concerns associated with alkaline electrolysis, they do not delve into the specifics of these risks or how they might be mitigated. In summary, while the manuscript provides an overview of the current state of tritium measurement techniques, it would be significantly strengthened by deeper analysis, clearer presentation of data, and a more forward-looking perspective on future research and applications.
Author Response
Comment 1: However, the originality of the research could be questioned as the manuscript largely summarizes previous work without presenting novel findings or approaches.
Response 1: Dear Reviewer, we thank you for raising the issue of originality in our manuscript. We acknowledge your concern and would like to address it as follows:
(1) The scarcity of literature on the online measurement of low-energy nuclides in water is indeed due to the niche nature of this research area and the technical challenges it presents. As such, there have been limited advancements in novel measurement methods to date.
(2) In response to this, we have revised the discussion section of our manuscript to provide a comprehensive summary of the current state of the technology. We have also included an
outlook on the future, discussing potential solutions to the existing problems and the directions in which the field might advance.
We believe that while our work may largely synthesize previous research, it serves to consolidate knowledge in an understudied area, offering a solid foundation for future studies. Our revised manuscript now better positions the work within the context of the field and underscores the need for further research.
Comment 2: While the review covers a broad range of techniques, it lacks in-depth analysis of the recent advancements in tritium detection technology.
Response 2: we appreciate your observation regarding the depth of analysis in our review, particularly concerning recent advancements in tritium detection technology. We understand the importance of thoroughly examining the latest developments in the field.
In response to your feedback, we have expanded our discussion on solid-state scintillators and added Table 5, comparing the material aspects of commonly used plastic scintillators. We have now included a more detailed examination of the latest materials used in this technology, comparing their performance metrics with those of traditional Liquid Scintillation Counting (LSC) methods. This additional analysis provides a clearer picture of the advantages and limitations of these new materials in the context of tritium detection.
Thank you for your insightful comments, which have significantly contributed to enhancing the quality and depth of our review.
Comment 3: Besides, the manuscript would benefit from the inclusion of more graphical data, such as charts or graphs, to illustrate the performance parameters of different techniques.
Response 3: Thank you for your suggestion to include more graphical data in our manuscript. We recognize the value of visual representation in effectively conveying complex information and aiding reader comprehension.
We have taken your advice and have now incorporated additional graphical abstract, charts and graphs including that illustrate the performance parameters of various tritium detection techniques. These visual aids compare key metrics such as sensitivity, detection limits, and response times, providing a clearer and more accessible overview of the comparative performance of different methods.
Your feedback has been instrumental in refining our manuscript, and we are grateful for the opportunity to make these important additions.
Comment 4: Moreover, the review should provide a more critical analysis of the limitations and potential pitfalls of each technique discussed.
Response 4: We greatly appreciate your valuable feedback on the need for a more critical analysis within our review. Your guidance has been instrumental in enhancing the depth and rigor of our discussion.
In response to your suggestion, we have expanded our discussion section to include a more critical examination of the limitations and potential pitfalls associated with each technique discussed in the review. We add some space to the text that specifically addresses the shortcomings of current technologies and offers insights into the future trends and possible advancements that may mitigate these issues.
Regarding the pre-treatment aspect, we acknowledge that the literature on this topic is relatively sparse. Consequently, our review has focused more heavily on the detection technologies themselves, where a more extensive body of research is available. However, we have made an effort to include any relevant findings on pre-treatment when discussing the overall performance and applicability of the detection methods.
Round 2
Reviewer 1 Report
Comments and Suggestions for Authors
Dear M. Junxiang et. al:
I am satisfied with your modifications, I recommend that the manuscript be accepted.
Author Response
Comment 1: I am satisfied with your modifications, I recommend that the manuscript be accepted.
Response 1:
Thank you for your positive endorsement and recommendation for the acceptance of our manuscript. We are honored by your support and are excited about the prospect of our research being included in the scholarly discourse.
We appreciate your time and expertise in reviewing our submission.
Reviewer 3 Report
Comments and Suggestions for Authors
The manuscript improved but still some minor revision remained.
The introduction section should has just one figure (Fig.2); Fig 1, Table 1, and all equations should not be in the introduction part.
Comments on the Quality of English LanguageNA
Author Response
Comment 1:The introduction section should has just one figure (Fig.2); Fig 1, Table 1, and all equations should not be in the introduction part.
Response 1: Thank you for your meticulous review and valuable suggestions. In response to your request, I have revised the introduction section to now only include Figure 2. Additionally, I have consolidated Figure 1, Table 1, and all necessary equations into Appendix A to ensure the coherence and completeness of the manuscript. I look forward to your further feedback and appreciate your support for our research.
Reviewer 4 Report
Comments and Suggestions for Authors
The high percentage of matching content suggests that the article may lack originality. It's crucial for academic and professional writing to present unique insights and perspective. Moreover, a significant match rate could indicate potential plagiarism, which is a serious ethical concern in any field. It's essential to ensure that all content is properly cited and that ideas are presented in your own words. Following are some suggestions that can help improve the work.
-The repetition might imply that the article does not delve deeply into the subject matter. Encouraging a more thorough analysis could help to reduce reliance on existing material.
-It's important to review the citation practices used in the article. Proper attribution should be given to all sources, and direct quotes should be used sparingly and appropriately.
-To reduce the match rate, consider expanding the argument with more in-depth research and a broader range of sources. This can help to present a more comprehensive view of the topic.
-If the repetition is due to paraphrasing, it may be beneficial to rewrite sections to ensure that the language is distinct and the ideas are conveyed in a novel way.
-If the article relies heavily on quotations, consider how these are integrated. It might be necessary to limit the use of direct quotes and instead summarize or synthesize information in your own words.
-The high repetition rate might be a sign that the thesis or central argument of the article is not clear or strong enough. A well-defined thesis can guide the writing process and help to reduce unnecessary repetition.
-It could be helpful to seek feedback from peers or mentors on the draft. Fresh perspectives can often identify areas of repetition and suggest ways to improve the originality of the content.
By doing the above, it is possible to create a piece of writing that is both informative and original, reflecting a deep understanding of the subject matter.
Author Response
Comment 1:The high percentage of matching content suggests that the article may lack originality. It's crucial for academic and professional writing to present unique insights and perspective. Moreover, a significant match rate could indicate potential plagiarism, which is a serious ethical concern in any field. It's essential to ensure that all content is properly cited and that ideas are presented in your own words.
Response 1:We greatly appreciate the valuable suggestions you provided for our manuscript. Your feedback has been carefully considered, and we have made the following revisions to reduce the high degree of content matching and to incorporate more of our perspectives:
1. Enhanced Originality: We have added a critical summary and analysis of the existing literature in Section 3, highlighting the strengths and limitations of current research.
2. Perspectives and Forecasts: Before and after the literature review, we have included our insights and judgments on future directions, based on current trends and challenges in the field.
2. Citation Practices: We have ensured proper attribution and integration of direct quotes, maintaining the originality and coherence of the manuscript.
4. Structural and Linguistic Improvements: The manuscript's structure and language have been refined for enhanced readability and logical flow.
We believe these revisions enhance the originality, academic contribution, and presentation quality of our work. We look forward to your further feedback and thank you for your support.